# Antimicrobial Activity of *Apis mellifera* Bee Venom Collected in Northern Peru

**DOI:** 10.3390/antibiotics12040779

**Published:** 2023-04-19

**Authors:** Orlando Pérez-Delgado, Abraham Omar Espinoza-Culupú, Elmer López-López

**Affiliations:** 1Health Science Research Laboratory, Universidad Señor de Sipán, Chiclayo 14001, Peru; 2Faculty of Health Sciences, Universidad Señor de Sipán, Chiclayo 14001, Peru

**Keywords:** venom, antimicrobial activity, fraction, bee, protein

## Abstract

Due to the emergence of microorganisms resistant to antibiotics and the failure of antibiotic therapies, there is an urgent need to search for new therapeutic options, as well as new molecules with antimicrobial potential. The objective of the present study was to evaluate the in vitro antibacterial activity of *Apis mellifera* venom collected in the beekeeping areas of the city of Lambayeque in northern Peru against *Escherichia coli*, *Pseudomonas aeruginosa,* and *Staphylococcus aureus*. Bee venom extraction was performed by electrical impulses and separated using the Amicon ultra centrifugal filter. Subsequently, the fractions were quantified by spectrometric 280 nm and evaluated under denaturant conditions in SDS-PAGE. The fractions were pitted against *Escherichia coli* ATCC 25922, *Staphylococcus aureus* ATCC 29213, and *Pseudomonas aeruginosa* ATCC 27853. A purified fraction (PF) of the venom of *A. mellifera* and three low molecular weight bands of 7 KDa, 6 KDa, and 5 KDa were identified that showed activity against *E. coli* with a MIC of 6.88 µg/mL, while for *P. aeruginosa* and *S. aureus,* it did not present a MIC. No hemolytic activity at a concentration lower than 15.6 µg/mL and no antioxidant activity. The venom of A. mellifera contains a potential presence of peptides and a predilection of antibacterial activity against *E. coli.*

## 1. Introduction

World Health Organization (WHO) published the first global surveillance report on antibiotic resistance, showing that five out of six regions had more than 50% resistance to third-generation cephalosporins and fluoroquinolones in *Escherichia coli* and methicillin resistance in *Staphylococcus aureus* in hospital settings [1].

*S. aureus* is a human commensal that can cause systemic infections in the host; this requires evading the immune response and the ability to proliferate in different niches in the host; currently, the infection by staphylococci in the face of immune mediators and the disease is not well known [2]. However, the main agent of bacteremia and infective endocarditis (IE), as well as osteoarticular, skin, and soft tissue infections, pleuropulmonary infections [3], and even the appearance of methicillin-resistant *S. aureus* (MRSA), which is a therapeutic problem in patients [4]. Staphylococcal infection has also been reported from hosts or carriers of asymptomatic nasopharyngeal bacteria, even with certain risk factors such as passive smoking and a large family [5]. The results of certain studies have determined that *S. aureus* has generated resistance against ampicillin, penicillin, rifampicin, clindamycin, oxacillin, and erythromycin [6]. Variable susceptibility to levofloxacin, ciprofloxacin, gentamicin, tetracycline, and sulfamethoxazole-trimethoprim has also been shown [7], and patterns have also been shown regarding the mecA, rpoB, blaZ, ermB, tetM, and nuc genes [6,8]. This resistance has been acquired through different mechanisms, the most frequent being reduced membrane permeability, excessive production of β-lactamase, and acquiring resistance genes or gene mutations [9].

Likewise, *Pseudomonas aeruginosa* can cause nosocomial outbreaks related to its resistance and virulence properties [10], being a producer of β-lactamases and multiresistant to a wide range of antimicrobials such as penicillin, cephalosporin, cephamycin, and carbapenem [11]. In addition, 35 resistomes (antimicrobial resistance genes) have been identified that confer resistance to 18 different antibiotics (including four classes of beta-lactams) and 214 virulence factor genes [12], in addition to the susceptibility of *P. aeruginosa* to carbapenems, piperacillin-tazobactam, and amikacin has undergone alterations before and during COVID-19 [13]. There are phenotypic studies that *P. aeruginosa*, as producers of extended-spectrum beta-lactamase (ESBL) and metallo-beta-lactamases (MBL), also present genes associated with biofilm formation and virulence, such as toxA and lasB [14]. Even using polymerase chain reaction (PCR) techniques and molecular markers such as Random Amplified Polymorphic DNA (RAPD), they have identified strains resistant to imipenem, Ticarcillin + Clavulanate, Piperacillin, and Ticarcillin + Clavulanate, these strains being isolated from swimming pools [15].

*E. coli* is responsible for a large number of virulent variants associated with human diseases, such as urinary tract infection (UTI) with a resistance rate of >55% to first to fourth-generation cephalosporins [16], neonatal and traveler’s diarrhea [17], and multiresistant isolates (MDRs) the most prevalent genes being CTX-M-1, followed by NDM-1 for Betalactamases and the genes ermB and aac(6′)-Ib for resistance to macrolides and aminoglycosides [18]. *E. coli* is frequently discharged into the environment through feces, including wastewater, and is considered an indicator of fecal contamination. Many strains can carry resistance genes [19]. According to isolates, they have usually been reported to be sensitive to netilmicin, gentamicin, chloramphenicol, pipemidic acid, nalidixic acid, ciprofloxacin, amoxicillin/clavulanic acid, and nitrofurantoin, as well as increased susceptibility to cefotaxime, ceftriaxone, and aztreonam [20]. Uncomplicated UTI isolates have been found to have a higher susceptibility than complicated UTI isolates to amoxicillin, amoxicillin/clavulanic acid, and ciprofloxacin [21].

Antimicrobial resistance has been reported in gram-negative and gram-positive bacteria, with reports of up to 96.2% for *Pseudomonas* spp. and 66.7% for *E. coli* [22], with isolates reported in heart disease intensive care units [23] and in bloodstream infections [24] antimicrobial-resistant bacteria are also considered the most frequent uropathogens [25]. There are reports of MRSA in up to 50% of patients [26], with resistance profiles for cefoxitin, chloramphenicol, clindamycin, and gentamicin [27].

The failure of antibiotics, including the latest generation, to counteract superbugs highlights the need to search for new molecules with antimicrobial potential to control the global problem of antimicrobial resistance. One group of these new molecules are peptides that are antimicrobial (AMP) and are promising molecules for combating antimicrobial resistance (AMR) [28]. AMP has been found the most in the venoms of different organisms, such as scorpions [29,30,31], snakes [32], spiders [33], and bees [34], among other venoms.

The composition of bee venom is very variable, having Peptides: Melitin (the main component of the venom), Apamin, Mast Cell-Degranulating Peptide (MCD), Tertiapin, Secapin, and Its Isoforms, Adolapin, Procamine, and Minimine; Polypeptides: Api m 6, Cardiopep, Icarapin, and Major Royal Jelly Proteins; Enzymes: Phospholipase A2 (PLA2), Hyaluronidase, Acid Phosphatase, and Dipeptidylpeptidase IV; Serine Proteases [35].

Bees are insects found on all continents, many of these species have yet to be described and are an exciting source for the study and search for new molecules with antimicrobial properties. There are experimental and clinical reports on *Apis mellifera* venom and its anti-inflammatory, antimicrobial, and anticancer effects; the components present in the venom, such as proteins, vary from a summer season compared to a winter season [36,37,38], in addition, have shown different therapeutic properties against oxidative stress induced by beta-amyloid [39,40,41]. For Parkinson’s disease, the neuroprotective potential of bee venom against oxidative stress induced by rotenone (pesticide) has been demonstrated in a mouse model, including preventing the decrease in dopamine and also restoring locomotor activity in mice [42,43]. For Lyme disease, the melittin present in the venom showed in vitro antibacterial effects against the causative agent *Borrelia burgdorferi* [44] and even had significant antibacterial effects against *E. coli*, *S. aureus*, and *Salmonella typhyimurium* [45]. Melittin also exhibited antibacterial activity against MRSA strains [46], with antimicrobial potential against agents that cause dental caries, with antifungal capacity including suppression of biofilm formation [47,48]. Its significant antiviral potential has also been demonstrated in in vitro and in vivo assays on different enveloped (Influenza A) and non-enveloped (enterovirus-71) viruses [49]. In addition, phospholipase A2 (PLA2) can also block the replication of the virus, being shown to be responsible for the inhibition of HIV replication [50]. The present study aimed to evaluate the in vitro antibacterial activity of *Apis mellifera* venom collected in the beekeeping areas of the city of Lambayeque in northern Peru against *Escherichia coli*, *Pseudomonas aeruginosa*, and *Staphylococcus aureus*.

## 2. Results

As seen in Figure 1, 15% SDS-PAGE-Tricine of the purified fraction (PF) of crude venom from *A. mellifera* yielded 3 low molecular weight bands, i.e., 7 kDa, 6 kDa, and 5 kDa.

The PF of *A. mellifera* venom had a minimum inhibitory concentration (MIC) of 6.88 µg/mL (*p* < 0.05) for *E. coli.* For the *S. aureus* strain, at higher concentrations, the venom exhibited antibacterial activity. For *P. aeruginosa,* no antibacterial activity was observed (Figure 2).

Concentrations of the PF of *A. mellifera* venom above 125 µg/mL in erythrocyte suspension produced more than 50% hemolysis. At a concentration of 31.25 µg /mL, less than 10% hemolysis occurred, but at 7.8 µg /mL PF, no erythrocyte lysis was evidenced (Figure 3).

The PF of the *A. mellifera* venom showed no antioxidant activity was observed at the concentrations evaluated.

## 3. Discussion

The purified fraction (PF) of the *A. mellifera* venom revealed the presence of peptides using the SDS-PAGE-Tricine technique and found three peptides of 7 kDa, 6 kDa, and 5 kDa. The chemical composition of the venom of *A. mellifera* is highly variable; such as melittin (3 kDa), apamin (2 kDa), and cecropin (4 kDa), enzymes, such as phospholipase A2 (19 kDa) and hyaluronidase (38 kDa), biologically active amines, such as histamine and epinephrine, as well as peptides not reported [39,51,52] and this suggests that many of these components may contribute their anti-inflammatory, antifungal, antiviral, healing and analgesic properties [36,53,54,55].

Our results were coherent with other studies; the PF of *A. mellifera* venom collected from Íllimo showed antibacterial activity against *E. coli*, but the same was not observed for *P. aeruginosa* despite being a gram-negative bacterium. For *S. aureus*, as the concentration of the venom fraction increased, bacterial growth was affected. Interestingly, the results of other studies of the antibacterial activity of the crude venom showed highly variable MICs against *E. coli* and for *S. aureus* [56], including also demonstrating the inhibitory effect through a viability assay at a temperature of 25 °C against *E. coli* and *P. putida*, causing membrane permeability and loss of ATP, showing no effect against *P. fluorescens* [57], as well as the crude bee venom extract in a region of Egypt significantly inhibited the growth of *E. coli* ATCC8739 and *S. aureus* ATCC 6538P [58], in addition, the action of crude bee venom from Iran demonstrated inhibition through the Kirby–Bauer method against *E. coli* and *S. aureus*, but not against *P. aeruginosa* [59].

Other studies have shown the existence of antimicrobial peptides in the venom; it follows that the interaction against the cell envelope of the bacteria is due to the attraction between the positively charged venom peptides and the phospholipids, causing a rupture or instability of the venom membrane, in addition to forming pores; however, this mechanism requires a certain concentration threshold [60]. Direct insertion of melittin leads to pore formation, whereas the parallel conformation is inactive and prevents other melittin molecules from being inserted, thus, preventing pore formation [61]. However, melittin has a molecular weight of 3 KDa; in our study, we found three peptides in the range of 5 KDa to 7 KDa; this finding demonstrates that melittin is not the only peptide present in bee venom with antibacterial activity.

The PF of *A. mellifera* venom at a concentration lower than 15.6 µg/mL demonstrated low hemolytic activity. Few studies have revealed the hemolytic activity of bee venom in Peru; on the contrary, in other latitudes, they have revealed that melittin has not presented significant hemolytic activity below a concentration of 0.25 µg/mL [62], the hemolytic action was also demonstrated against erythrocytes of different species, with variable sensitivity to bee venom pools, with sheep erythrocytes being the most resistant to hemolytic action compared to equine erythrocytes, including humans erythrocytes showed good resistance to hemolytic action, it follows that hemolysis can be increased by the action of phospholipase 2 (PLA2) after the action of melittin [63]. 

The PF of the venom of *A. mellifera* has not shown antioxidant capacity, but in other studies, they demonstrated antioxidant capacity. Curiously, they worked with the total venom or apitoxin, having the capacity to inhibit the free radical DPPH (2,2-diphenyl-1-picrylhydrazine) between 60% and 75% of antioxidant activity [64]. In the same sense, demonstrated with the venom of *A. mellifera syriaca* eliminating DPPH radicals between 50 to 65% [65]. When analyzing the venom of four bee species, *A. dorsata*, *A. mellifera*, *A. florea*, and *A. cerena*, they showed that *A. dorsata* contained the highest amount of melittin; they also revealed that the extract of *A. dorsata* had the highest antioxidant activity from the DPPH and ABTS (3-ethylbenzothiazoline-6-sulfonic acid) assays, including melittin alone, revealed very poor antioxidant activity among all bee venom extracts [66], this suggests that in our study, of the peptides present in the venom PF, melittin was not present.

The bioactive components present in the venom have generated much interest in medicine through the different species of the *Apis* genus, and their application in in vitro antimicrobial activity [67], their cytotoxic action against cancer cells [68], even the synergistic effect of the venom with some antibiotics such as Cephotax, Cefepime, and Tavanic has been revealed [69]. Through Transmission Electron Microscopy, the deformation of the cell wall was appreciated, resulting in the destruction of the cell wall, changes in the permeability of the membrane, leakage of cell content, inactivation of metabolic activity, and finally, cell death [57], as the inhibitory effect on F1-F0—ATPase has also been demonstrated [70].

## 4. Materials and Methods

### 4.1. Bee Venom Samples

The venom was obtained from Africanized bees *A. mellifera* (Linnaeus, 1758) in hives from the Cruz Verde town center of the Ilimo District located at latitude 6°28′26″ and longitude 79°50′34″ (Lambayeque); an electrical impulse of 3 volts with an electrical intensity of 0.004 A was passed through a collecting box (beeWhisper 6.0; Model 2020), without harming the specimens (Figure 4a,b) [71]. 

The venom was collected on a glass plate and allowed to dry; it was then transferred into a 50-mL Falcon tube (Figure 4c,d). Then, the bee venom was resuspended in sterile deionized water and centrifuged at 10,000× *g* at 4 °C for 10 min to remove insoluble materials. The supernatant was collected and stored in 2 mL microtubes at −20 °C.

### 4.2. Fraction Concentration and Electrophoresis 

The fractions were collected and quantified via absorbance at 280 nm (Navi UV/vis Nano spectrophotometer, Seongnam-si, Republic of Korea) using the formula [mg/mL] = (1.56 × Abs 280 nm) − (0.76 × Abs 260 nm) [72].

Crude fractions of bee venom were collected and concentrated using the Amicon ultra centrifugal filter (Merck Millipore, Cork, Ireland) with Cutoff from 3 kDa to 100 kDa [73], quantified via absorbance, and 50 µg protein from the FP were evaluated on a gel Tricine-SDS-PAGE (15%) under denaturing conditions [74] with a voltage of 100 volts and stained with Coomassie blue [75].

### 4.3. Antimicrobial Activity Test

The MIC values of the fraction were determined using the broth microdilution method in 96-well plates [76] against the strains of *E. coli* ATCC 25922, *S. aureus* ATCC 29213, and *P. aeruginosa* ATCC 27853; 50 µL of bacterial solution containing 5 × 10^4^ CFU/mL was placed in each well, then 50 µL of different concentrations of the fraction (55 µg/mL to 3.44 µg/mL) were added and incubated at 37 °C for 24 h. The positive control was broth plus inoculum, and the negative control was only broth. Growth of the positive control was determined by a growth button of ≥2 mm or defined turbidity. Finally, the plates were read by absorbance at 630 nm (SmartReader 96—Accuris) to determine the minimum inhibitory concentration (MIC). All assays were performed in triplicate.

### 4.4. Evaluation of Hemolytic Activity

The hemolytic test was evaluated following the protocol described by Oddo [77], red blood cells washed with PBS (137 mM NaCl, 2.7 mM KCl, 10 mM Na_2_HPO_4_, 1.8 mM KH_2_PO_4_) and resuspended at a concentration of 0.5% and incubated for 1 h with different concentrations of the fractions and then centrifuged at 10,000× *g* for 10 min, 60 µL of supernatant was transferred to a 96-well polypropylene plate, and the absorbance was read at 405 nm. The results were normalized with the positive controls of hemolysis (0.25% SDS) and negative controls (PBS). Assays were performed in triplicate.

### 4.5. Evaluation of Antioxidant Activity

20 μL of different concentrations of the fraction (55 µg/mL to 3.44 µg/mL) were added with 380 μL of ABTS radical in ethanol, incubated at room temperature protected from light for 30 min, then the absorbance of the mixture was measured at 734 nm [66]. To calculate the % decoloration, the following equation was used: % decoloration = [(C − S)/C] × 100, where C is the absorbance of the control, and S is the absorbance of the problem sample. Trolox was used as a positive control. The experiments were done in triplicate.

### 4.6. Statistic Analysis

The MegaStat add-in for Excel was used to determine the antibacterial activity of the purified fraction of *Apis mellifera* venom. Analysis of variance (ANOVA) was performed at a significance level of 5%.

## 5. Conclusions

In summary, *A. mellifera* bee venom contained peptides with weights of 7 kDa, 6 kDa, and 5 kDa and exhibited antibacterial activity against *E. coli* ATCC 25922 at a concentration of 6.88 µg/mL.

## Figures and Tables

**Figure 1 antibiotics-12-00779-f001:**
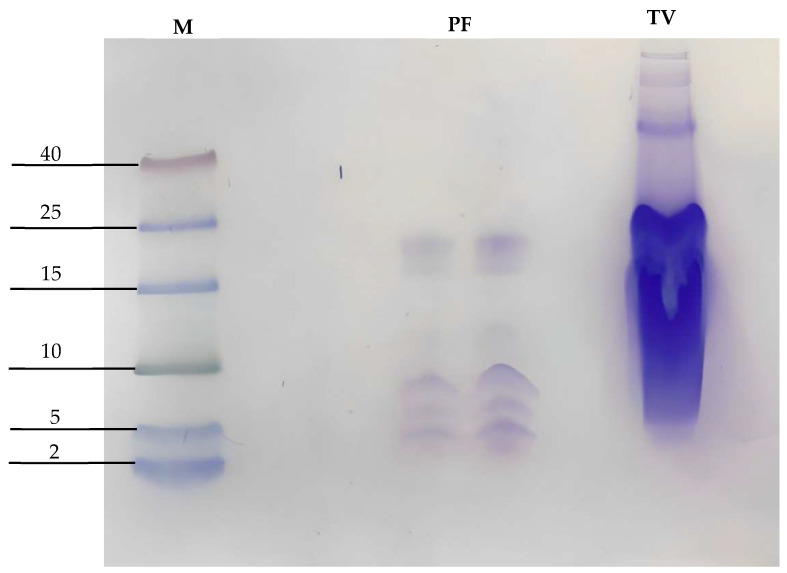
SDS polyacrylamide gel electrophoresis with Tricine (SDS-PAGE-Tricine) of the purified fraction; M denoted the marker lane (molecular weight 2–40 kDa) (molecular weight marker), PF denotes the protein fractions, and TV denotes total venom.

**Figure 2 antibiotics-12-00779-f002:**
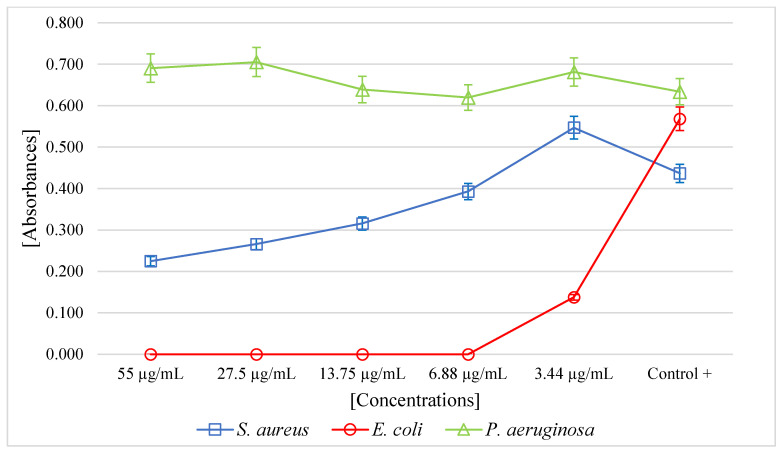
Microbial growth of *S. aureus*, *E. coli*, and *P. aeruginosa* incubated with the purified fraction of *Apis mellifera* venom.

**Figure 3 antibiotics-12-00779-f003:**
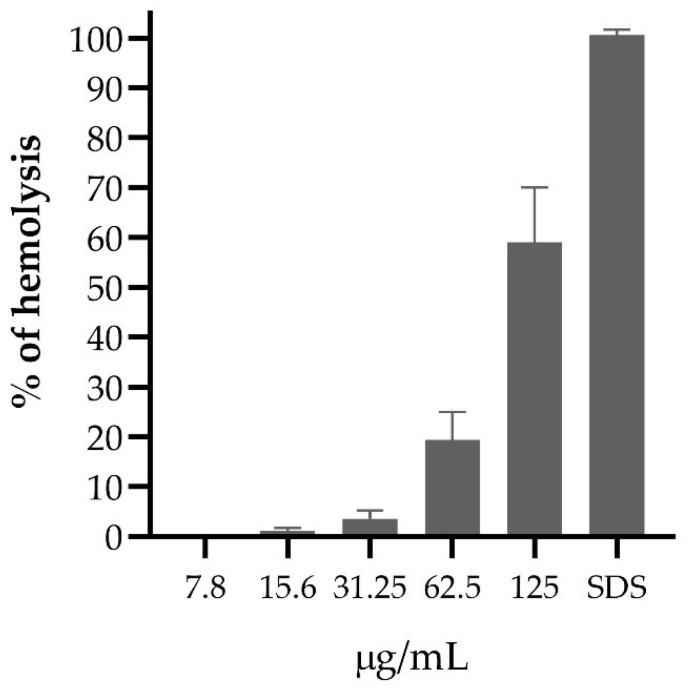
Hemolytic activity of the purified fraction of *Apis mellifera* venom.

**Figure 4 antibiotics-12-00779-f004:**
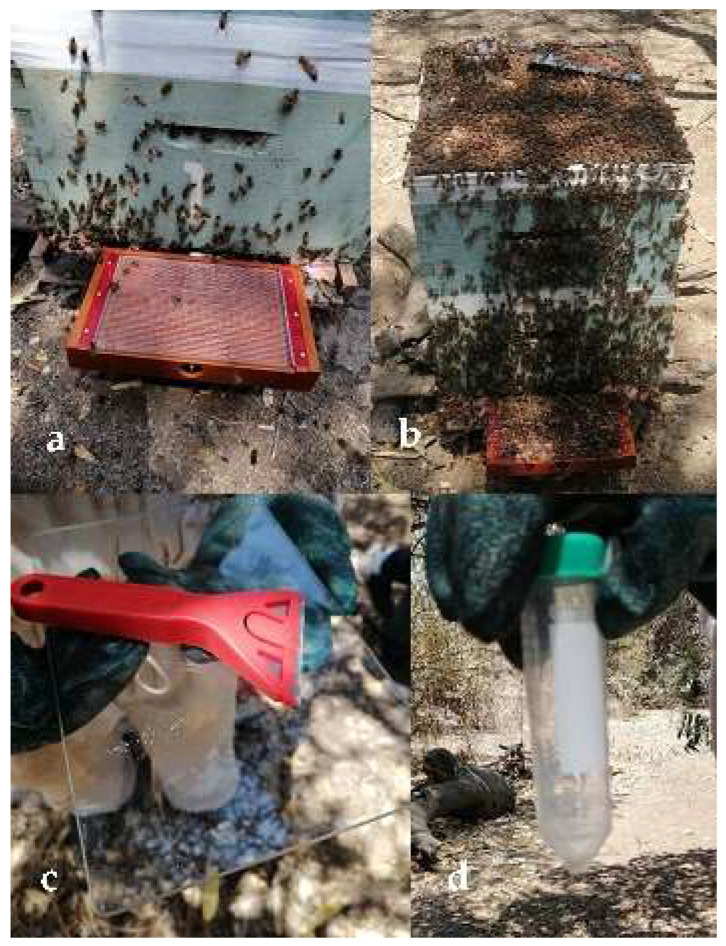
(**a**) beeWhisper 6.0. collector box; (**b**) electrostimulation of the collector box; (**c**) recovery of *A. mellifera* dry venom; (**d**) storage of the venom.

## Data Availability

All data related to the manuscript were available in the main manuscript.

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
