# Peer review of "Antimicrobial Activity of *Apis mellifera* Bee Venom Collected in Northern Peru"

_antibiotics, 2023, doi:10.3390/antibiotics12040779_

Round 1
Reviewer 1 Report
What is new about this work?
The bacterial strains use in the research are very common ..please improve the experimental design..
How you used the bee venom in the research because in solid form is impossible. and in the solution can precipitate?
The methods are summarily described, also the conclusions. and discussions regarding the research are missing...
Author Response
What is new about this work?
It is the first work carried out in Peru with bee venom and partial isolation of peptides with antibacterial capacity, three different peptides than Melitin were found.
The bacterial strains use in the research are very common ..please improve the experimental design.
Standard strains were used, based on this preliminary result it can be increased with new gram-positive and gram-negative bacteria, fungi, among other microorganisms.
How you used the bee venom in the research because in solid form is impossible. and in the solution can precipitate?
The solid venom was resuspended in deionized water, refrigerated centrifugation and the amicon system were used to concentrate the peptides, there was no evidence of precipitation in the assay.
The methods are summarily described, also the conclusions. and discussions regarding the research are missing
They are standardized methods for antimicrobial testing and even hemolysis and antioxidant tests were performed to ensure that the components found in the venom are of pharmacological interest.
The discussion, an analysis of the results with previous works has been carried out.
Reviewer 2 Report
Please see attached comments

Author Response
Changes were made to the manuscript
Introduction:
- A new paragraph has been added (Lines 26 to 29)
- New citations have been incorporated: Line 40, Line 42, (antibiotics have been referenced) Line 43 and 45 (on the mechanisms of acquisition of resistance to antibiotics)
- A new paragraph has been incorporated, Lines 79 to 85, 86 to 91 on the composition of bee venom.
- A new paragraph Lines 92 to 94 has been incorporated.
Results:
- MIC acronyms were defined
- The highest concentration was 55 µg/mL
Discussion:
- Discussion reworded
Materials and methods:
Added word africanized (line 249)
Reviewer 3 Report
The study proposed by the authors presents an interesting area of alternative sources for antibioticts, which causes interest for both scientists and practitioners.
The authors, no doubt, did a good job, including the application of natural products such as bee venom and using traditional analysis methods in this research.
While reading the article, one important remark arised, by answering which the authors will improve the presentation of the results: figures 2 and 3 (2. discussions) require clarification and a more in-depth discussion. The current discussion is laconic and not insightful, that wouldn’t be understood by average reader.
Author Response
Figures 2 and 3 (2. discussions) require further clarification and discussion. The current discussion is laconic and not insightful, that would not be understood by the average reader.
Answer:
The discussion of the manuscript has been reformulated
Round 2
Reviewer 1 Report
In the introduction, please reformulated all...highlighting the role of bee venom, the properties for which this bee product can be studied, and their antimicrobial activities in recent studies.
Please improve your experimental design, because the number of samples and analyses doesn t relevant.
Author Response
In the experimental design, it is evidenced in the methodology and the results, to describe the research, 5 different concentrations of the purified fraction were used, positive and negative control, in triplicate with three different bacterial strains, having 63 experimental units for antimicrobial activity in vitro, 18 experimental units for hemolytic activity and antioxidant activity)
Between lines 309 to 311, it was included on the controls
Reviewer 2 Report
Thank you to all the authors for your hard work and for the revisions.
Author Response
Thank you so much for the comments